# Finding a right place to cut: How katanin is targeted to cellular severing sites

Masayoshi Nakamura[1] , Noriyoshi Yagi[1] and Takashi Hashimoto[2]

[1]Institute of Transformative Bio-Molecules (WPI-ITbM), Nagoya University, Nagoya, Japan; [2]Division of Biological Sciences, Nara Institute of Science and Technology, Ikoma, Japan

katanin; microtubule severing; plant cells.

**Authors for correspondence:**
M. Nakamura and T. Hashimoto,
E-mail: mnakamu@itbm.nagoya-u.ac.jp,
hasimoto@bs.naist.jp

## Abstract

Microtubule severing by katanin plays key roles in generating various array patterns of dynamic microtubules, while also responding to developmental and environmental stimuli. Quantitative imaging and molecular genetic analyses have uncovered that dysfunction of microtubule severing in plant cells leads to defects in anisotropic growth, division and other cell processes. Katanin is targeted to several subcellular severing sites. Intersections of two crossing cortical microtubules attract katanin, possibly by using local lattice deformation as a landmark. Cortical microtubule nucleation sites on preexisting microtubules are targeted for katanin-mediated severing. An evolutionary conserved microtubule anchoring complex not only stabilises the nucleated site, but also subsequently recruits katanin for timely release of a daughter microtubule. During cytokinesis, phragmoplast microtubules are severed at distal zones by katanin, which is tethered there by plant-specific microtubule-associated proteins. Recruitment and activation of katanin are essential for maintenance and reorganisation of plant microtubule arrays.

## 1. Introduction

Microtubules are dynamic hollow polymers consisting of 13 straight protofilaments formed by the head-to-tail polymerisation of $\alpha,\beta$-tubulin heterodimers. Microtubules become organised to form various arrays to perform suitable cellular functions in need for particular cell types and during cell cycle, including chromosome segregation, cell polarity and cell morphogenesis (Kirschner & Mitchison, 1986). In plant interphase cells, plasma membrane-associated cortical microtubules guide polar deposition of cellulose microfibrils in newly formed cell walls (Paredez et al., 2006), and contribute to straight elongation of axial organs, formation of complex cell shapes and development of secondary cell wall patterns (Ehrhardt & Shaw, 2006; Lin & Yang, 2020; Oda & Fukuda, 2012). Microtubule array may quickly adopt a new pattern or become transiently disassembled in response to external and internal cues, such as light, osmotic stress and phytohormones (Fujita et al., 2013; Lindeboom et al., 2013; Shibaoka, 1994). Generation, disassembly and maintenance of diverse microtubule arrays are generally governed by microtubule nucleation, microtubule regulators and microtubule-based motor proteins (Hamada, 2014; Nakamura, 2014; Nebenführ & Dixit, 2018). Microtubule severing also greatly contributes to (re)organisation of plant microtubules. In this review, we update previous reviews on plant katanin (Luptovčiak et al., 2017; Nakamura, 2014) and discuss how katanin is targeted to distinct subcellular locations.

## 2. Katanin severs microtubules

Among related eukaryotic enzymes that generate internal breaks in microtubules, katanin is the sole severing enzyme established in plants (McNally & Roll-Mecak, 2018). Katanin consists of a p60 catalytic subunit and a p80 regulatory subunit (Hartman et al., 1998). The p60 subunit belongs to the ATPase associated with diverse cellular activities (AAA)-type ATPase super-family and consists of an N-terminal microtubule interacting and trafficking domain and a C-terminal AAA ATPase domain, connected by a disordered linker region. Microtubule promotes

hexamerisation of the catalytic subunit, and stimulates ATPase and severing activities. Katanin hexamer forms a flat spiral ring, and pulls the tubulin C-terminal tail into the central pore. The electronegative multiple glutamates in the human $\beta$-tubulin are critical for binding and severing of microtubules (Zehr et al., 2020); plant $\beta$-tubulins are also rich in glutamates at their C-terminal tails. Microtubule is thought to break in the middle when sufficient numbers of tubulin heterodimers have been removed. Live imaging of *Arabidopsis* leaf epidermal cells suggests that several katanin hexamers need to accumulate at the severing sites before microtubule breakage occurs (Yagi et al., 2021).

Unlike mammals, which possess more than one katanin p60 genes (Lynn et al., 2021), a single p60 gene is found in the *Arabidopsis* genome. Several genetic screens targeted for distinct phenotypes resulted in various allele names for *Arabidopsis* katanin mutants (Luptovčiak et al., 2017). The p60 katanin mutants display disorganisation of cortical microtubule arrays and are defective in array reorganisation upon stimuli, which result in aberrant deposition of cellulose microfibrils, decreased cell wall strength, reduction in anisotropic cell expansion and a local decrease in growth heterogeneity (Bichet et al., 2001; Bouquin et al., 2003; Burk et al., 2001; Lindeboom et al., 2013; Sassi et al., 2014; Uyttewaal et al., 2012). Genetic screening also revealed that katanin-mediated microtubule dynamics plays a role in miRNA-guided translational repression (Brodersen et al., 2008).

The WD-40 repeat-containing p80 regulatory subunit promotes formation of a stable heterodimeric complex (Zehr et al., 2017), and is essential for targeting of katanin to specific cellular severing sites, such as centrosomes (Hartman et al., 1998). The interface formed by p60 and p80 is recognised by animal katanin regulators that recruit katanin to the growing minus ends of microtubules (Jiang et al., 2017; Jiang et al., 2018). In *Arabidopsis* leaf epidermal cells of the p60 catalytic mutant, the p80 subunit by itself is still targeted to the normal subcellular severing sites, indicating that the regulatory subunit mediates subcellular targeting (Wang et al., 2017). *Arabidopsis* possesses four p80 regulatory subunit genes. The quadruple p80 mutant shows growth and morphological phenotypes reminiscent of the p60 mutants (Wang et al., 2017). Although they are largely redundant, it is possible that these subunits confer partly distinct regulatory roles to the p60–p80 complex in plant cells.

Recombinant *Arabidopsis* p60 subunit fragments Taxol-stabilised microtubules into shorter lengths (e.g., Burk & Ye, 2002). Its overexpression in *Arabidopsis* causes similar fragmentation of cortical microtubules (Stoppin-Mellet et al., 2006), suggesting that under such conditions, katanin severs microtubules along their length. However, in wild-type plant cells, katanin seldom severs microtubules along the entire lattice, but instead is recruited to select subcellular locations, owing to the regulatory functions of the p80 subunit (Wang et al., 2017). Lateral cross-linking of adjacent microtubules by the microtubule-bundling protein MAP65-1 inhibits the binding of katanin along the sidewalls of microtubules in vitro (Burkart & Dixit, 2019). The bundled microtubules may thus be inefficient substrates for katanin-mediated severing in plant cells. Major cellular severing sites by katanin are described below (Figure 1).

## 3. Crossover sites of interphase microtubules

In interphase plant cells, cortical microtubules are associated with the inner surface of the plasma membrane along their length, and migrate the cell cortex by a hybrid treadmilling mechanism (Shaw et al., 2003). Such dynamic microtubules frequently collide each other, thereby promoting ordering of the cortical array (Dixit & Cyr, 2004). At the microtubule crossover sites, katanin accumulates and severs the crossing (or overlying) microtubules, rather than the crossed (or underlying) ones (Lindeboom et al., 2013; Wightman & Turner, 2007; Zhang et al., 2013). New plus ends of severed microtubules shrink rapidly in vitro (Walker et al., 1989). In *Arabidopsis* hypocotyl epidermal cells, the lagging halves of the severed microtubules are largely depolymerised from the exposed plus ends (Zhang et al., 2013). However, upon illumination with blue light, new plus ends of severed microtubules are protected from depolymerisation, largely by the action of plus-end stabilising protein CLASP, and resume growing (Lindeboom et al., 2019). Regrowth of severed microtubules amplifies the number of discordant microtubules and eventually reorients whole microtubule arrays to a new direction, which promotes curvature in response to light stimulus (Lindeboom et al., 2013). Katanin-mediated severing events also increase microtubule mass in non-plant organisms, such as at the meiotic spindle *in Caenorhabditis elegans* (Srayko et al., 2006), indicating microtubule amplification by severing may be used in non-centrosomal microtubule arrays (Roll-Mecak & Vale, 2006). In silico simulations indicate that preferential severing of crossing microtubules at crossovers strongly enhances array alignment (Deinum et al., 2017).

How katanin recognises and is recruited to the microtubule crossovers? Short stretches of GTP-bound tubulins form a protecting GTP cap at the growing plus end of microtubules, while the main body of the microtubule shaft is composed of GDP-tubulins (Mitchison & Kirschner, 1984). When dynamic microtubules were attached on coverslips and were set to collide each other in vitro, as observed for plant cortical microtubules, small GTP-tubulin patches accumulated at microtubule crossovers (de Forges et al., 2016). In animal PtK2 cells, free tubulins (presumably in a GTP-bound form) were found to be incorporated to the microtubule lattice preferentially at crossovers (Aumeier et al., 2016). Free tubulin incorporation along the shafts of microtubules results from local microtubule damage and subsequent repair (Schaedel et al., 2015). Since katanin has a strong affinity toward sites of damaged microtubule lattice in vitro (Davis et al., 2002; Diaz-Valencia et al., 2011), microtubule crossovers may be prone to local defects caused by geometric constraints and likely provide intrinsically high-affinity katanin binding sites (Figure 2). In the quasi-2D plane of plant cell cortex, crossing microtubules may experience more severe lattice deformation than crossed microtubules, thus leading to higher severing frequencies. Although local lattice defects intrinsically promote katanin recruitment at crossovers, this does not exclude elusive involvement of specific katanin targeting factors at these sites.

Microtubule severing probability at crossovers is influenced by factors other than katanin activities. SPIRAL2 is a minus-end tracking and stabilising protein (Fan et al., 2018; Leong et al., 2018; Nakamura et al., 2018). Suppressed minus-end depolymerisation of crossing microtubules increases lifetime of microtubule crossovers, resulting in a greater opportunity time for severing by katanin (Fan et al., 2018; Nakamura et al., 2018). Although SPIRAL2 forms dynamic aggregates at crossovers, it remains controversial whether this microtubule regulator directly affects severing (Nakamura et al., 2018; Wightman et al., 2013). A fraction of AUG3, a conserved subunit of augmin (see below), is also localised at microtubule crossovers (Wang et al., 2018), but its potential roles there need further investigation.

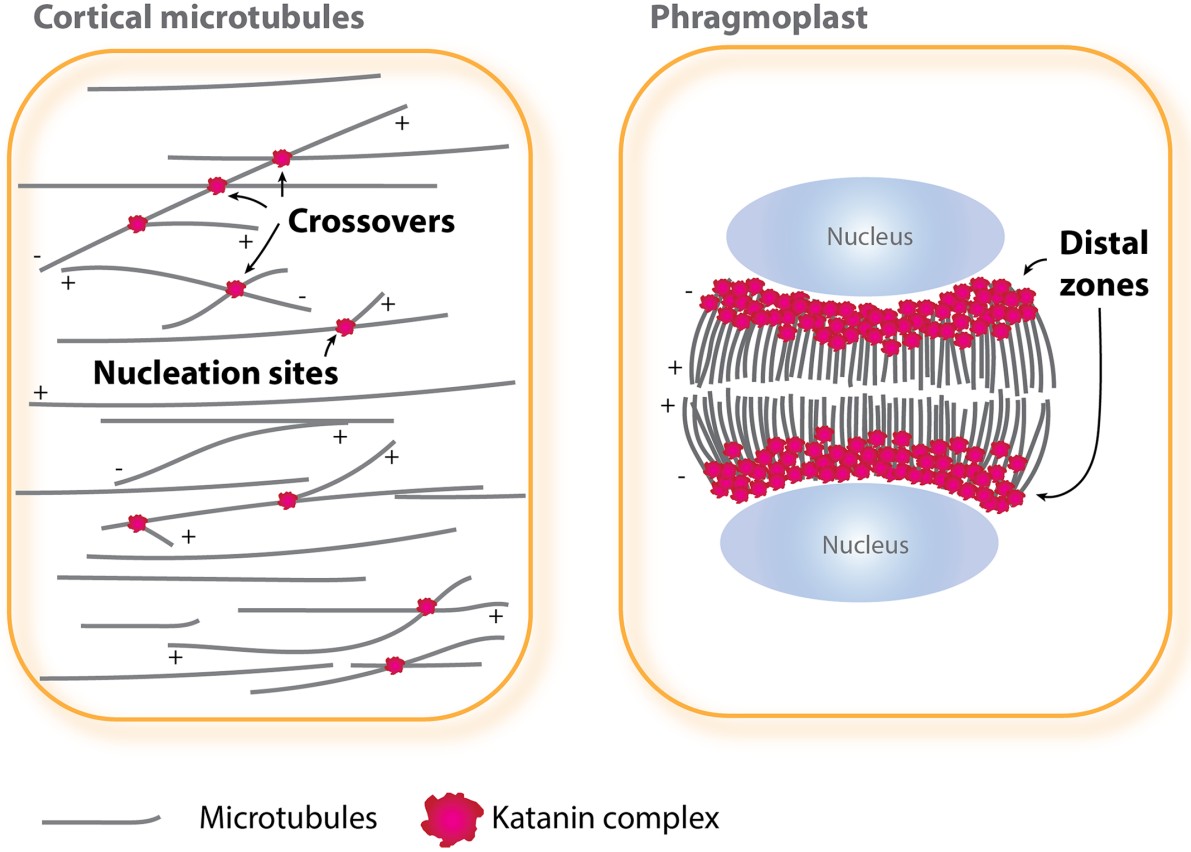

**Fig. 1.** Katanin-mediated microtubule severing in interphase and cytokinesis. In interphase cells (left), katanin is localised at and severs nucleation sites and crossover sites of cortical microtubules, whereas in cytokinetic cells (right), katanin functions at a distal zone of expanding phragmoplasts. The plus (+) and minus (−) ends of microtubules are indicated.

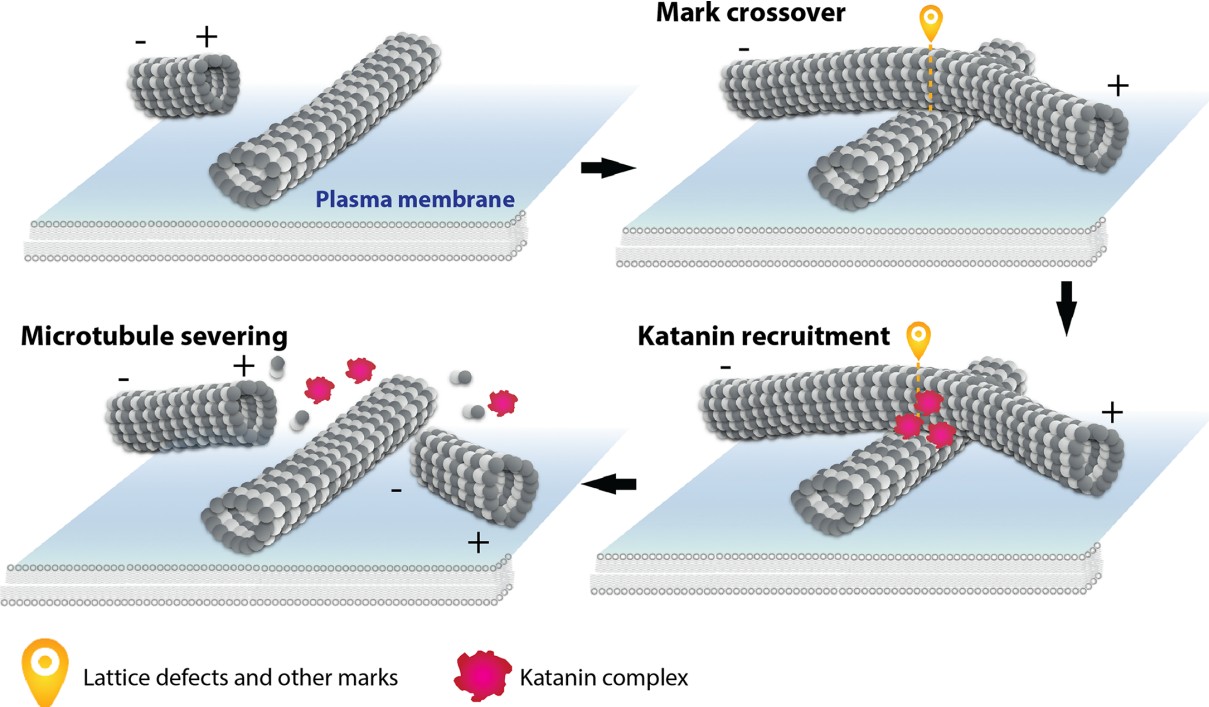

**Fig. 2.** Katanin-mediated microtubule severing at microtubule crossover sites. In plant interphase cells, crossovers are generated when two plasma membrane-associated cortical microtubules collide. Katanin accumulates at the crossover site by recognising elusive marks, possibly involving lattice defects, and severs an overriding microtubule. Since the precise geometry of microtubule crossover sites has not been clarified at the nanoscale level, the presented images are speculative and serve only for mechanistic discussions. The plus (+) and minus (−) ends of microtubules are indicated.

## 4. Nucleation sites of interphase microtubules

Microtubules in eukaryotic cells are generally nucleated from ~2.2 MDa $\gamma$-tubulin complexes composed of $\gamma$-tubulin, related $\gamma$-tubulin complex proteins and a few other proteins (Thawani & Petry, 2021). The eight-subunit protein complex augmin recruits $\gamma$-tubulin complexes to the microtubule lattice to initiate microtubule-dependent microtubule nucleation (Song et al., 2018). In acentrosomal plant cells, microtubules are nucleated from $\gamma$-tubulin complexes mainly dispersed on the lattice of preexisting microtubules (Ehrhardt & Shaw, 2006; Murata et al., 2005; Nakamura et al., 2010) in an augmin-dependent manner (Lee et al., 2017; Liu et al., 2014).

In interphase cortical arrays, microtubule-localised $\gamma$-tubulin complexes generate new (daughter) microtubules either at a branching angle of ~40° or in a parallel fashion, which results in immediate bundling with the mother microtubules (Chan et al., 2009; Murata et al., 2005; Yagi et al., 2018). The daughter microtubules are subsequently detached at the nucleation sites by katanin activities, and this nucleation-and-severing process is crucial for cortical microtubule array organisation and reorientation (Lindeboom et al., 2013; Zhang et al., 2013). In *katanin* mutant cells, daughter microtubules are not released from the nucleation sites and the $\gamma$-tubulin complexes remain attached at the nucleation sites (Nakamura et al., 2010). Katanin-mediated severing thus recycles once nucleated $\gamma$-tubulin complexes to a cytoplasmic pool for another round of nucleation.

Katanin is recruited to cortical microtubule nucleation sites by an evolutionarily conserved tethering complex composed of mitotic spindle disanchored 1 (Msd1) and WD repeat-containing protein 8 (Wdr8). The Msd1–Wdr8 complex was initially discovered as a critical factor that anchors microtubule minus ends to spindle pole bodies, the centrosome equivalent in fission yeast, partly by interaction with $\gamma$-tubulin complexes (Toya et al., 2007; Yukawa et al., 2015). In vertebrate cells, this complex tethers microtubules to the centrosome (Hori et al., 2015). In *Arabidopsis*, Msd1 was first identified as a microtubule-associated protein (MAP), whereas Wdr8 was efficiently recovered in the MAP fractions but does not itself bind microtubules (Hamada et al., 2013). Recently, these two proteins were found to form complexes in vivo, and to be recruited to the cortical microtubule nucleation sites immediately following recruitment of $\gamma$-tubulin complexes there (Yagi et al., 2021; Figure 3). The complex strongly stabilises the branching nucleation sites; its absence leads to facile dissociation of the minus ends of daughter microtubules even without katanin activities. Remarkably, katanin recruitment to the cortical microtubule nucleation sites absolutely requires the Msd1–Wdr8 complex at these locations. Katanin recruitment to microtubule crossovers (see above) does not depend on this tethering complex, however. Thus, the plant Msd1–Wdr8 complex possesses dual functions; it tethers the minus ends of daughter microtubules to the nucleated $\gamma$-tubulin complexes on the mother microtubules, thereby stabilising the branched microtubule structures, while it also provides a katanin-recruitment site for subsequent severing and release of daughter microtubules. Such antagonising functions of Msd1–Wdr8 are thought to place the release of daughter microtubules under the strict control of katanin activities.

## 5. Microtubules at cell division

Mitotic spindle and the cytokinetic phragmoplast are bi-polar microtubule assemblies in which abundant microtubules are aligned in a roughly parallel fashion with their plus ends facing toward the midzone. Microtubule-dependent microtubule nucleation occurs in these arrays (e.g., Murata et al., 2013) and requires $\gamma$-tubulin complexes and augmin (Lee & Liu, 2019), but high microtubule densities make detailed cell biological analyses on nucleation and release of daughter microtubules challenging. Microtubule nucleation may be regulated differentially between interphase and mitotic cells, as suggested by distinct subunit compositions of augmin in interphase and mitosis (Lee et al., 2017).

Katanin localises to mitotic microtubules arrays. Loss-of-function mutants of katanin p60 display misorientation of the mitotic plane (Bichet et al., 2001; Panteris et al., 2011; Webb et al., 2002), a delay in spindle positioning, and prolonged mitotic duration (Komis et al., 2017). The shape of the phragmoplast is affected, and centrifugal expansion of phragmoplast is delayed (Komis et al., 2017; Panteris et al., 2011). Whether katanin localises to nucleation sites, crossover sites or other sites of mitotic microtubules has not been critically determined due to dense array organisation. Subcellular localisation of Wdr8 largely overlaps with that of katanin on mitotic microtubules, but the Msd1–Wdr8 complex is not required for mitotic microtubule functions (Yagi et al., 2021), indicative of distinct regulation of katanin recruitment in mitotic cells. CORTICAL MICROTUBULE DISORDERING 4 (CORD4) and CORD5 are mitosis-expressed members of plant-specific MAPs (Sasaki et al., 2019). These CORD proteins accumulate and co-localise with katanin at the distal phragmoplast zone. In the *cord4* and *cord5* mutant cells, katanin localisation expands to entire phragmoplast microtubules, which results in abnormally long and oblique phragmoplast microtubules and slow expansion of phragmoplasts. Ectopic expression of CORD4 in interphase plant cells caused recruitment of the katanin p80 regulatory subunit to cortical microtubules and their fragmentation in a katanin-dependent manner (Sasaki et al., 2019). These results suggest that mitotic CORD proteins target katanin to the distal phragmoplast location (Figure 1), thereby promoting phragmoplast expansion by localised microtubule severing.

After cytokinesis, microtubules are temporally nucleated from the edge of newly formed cell walls, and the surface of nuclear envelope (Fishel & Dixit, 2013). Although these microtubules may be released and incorporated into reorganised arrays, it is not known whether katanin is involved in these processes.

## 6. Outlook

Cellular katanin functions can be regulated by multiple levels, such as protein abundance, cellular localisation, microtubule binding and microtubule-severing activity. Blue light-stimulated and phototropin-dependent microtubule severing in plant cells (see above; Lindeboom et al., 2013) is an excellent example of environmental control of katanin functions. Studies in animal systems reveal that phosphorylation and ubiquitylation of katanin are particularly important regulatory mechanisms (Lynn et al., 2021). Since katanin hexamers assembled in vitro from recombinant p60 subunit show robust severing activities, phosphorylation may generally function as inhibitory modifications. The katanin hexamer structure complexed with a tubulin C-terminus-mimicking peptide provides structural insight into how some phosphorylated katanin residues inhibit ATPase activity of katanin (Zehr et al., 2020). Several phosphorylation sites have been reported for plant katanin (e.g., Plant PTM Viewer; https://www.psb.ugent.be/webtools/ptm-viewer/index.php). Functional consequences of these phosphorylation sites need to be

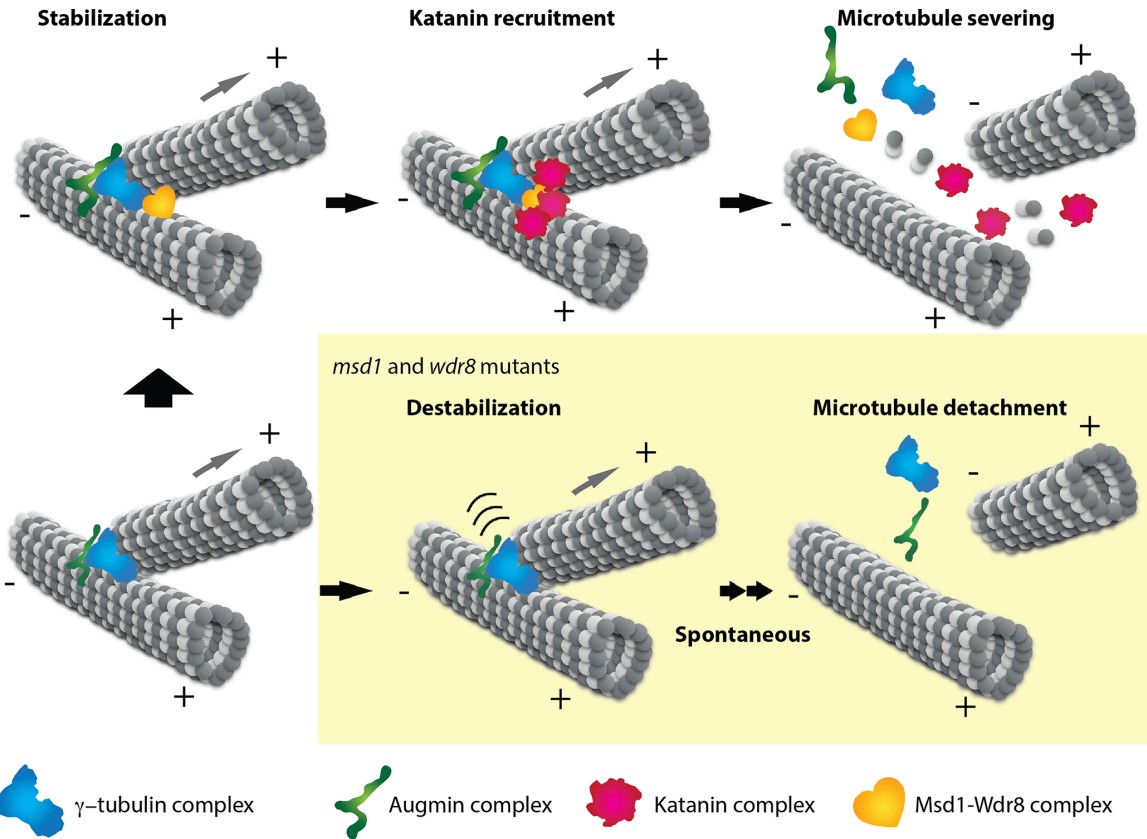

**Fig. 3.** Katanin-mediated severing and release of a daughter microtubule after nucleation. In plant interphase cells, both the γ-tubulin complex and the augmin complex accumulate at the lattice of existing cortical microtubules, and nucleate a daughter microtubule. After nucleation, the Msd1–Wdr8 complex is required to stabilise the branching nucleation structure; its absence results in spontaneous detachment of a daughter microtubule (yellow panel). The Msd1–Wdr8 stabilising complex subsequently recruits katanin for regulated severing and release of a daughter microtubule. The plus (+) and minus (−) ends of microtubules are indicated.

addressed in the future. Rho GTPase signalling pathway is another candidate for katanin regulation (Lin et al., 2013).

Animal tubulin tails are posttranslationally modified with multiple glutamates, which are gripped by the katanin central pore and enhance severing activity (Zehr et al., 2020). Thus, glutamylation levels of tubulins provide potential regulatory mechanisms for katanin activity. However, plant tubulins show no evidence of posttranslational modification by glutamylation (Hotta et al., 2016), making tubulin glutamylation an unlikely regulation for plant katanin.

In this review, we described katanin as an enzyme that severs a microtubule into two fragments or detaches a daughter microtubule from the nucleation site. Interestingly, a recent in vitro reconstitution study demonstrated that worm katanin can partially damage the microtubule shaft by extracting several molecules of tubulin heterodimers, and the nanoscale damage sites are actively repaired by incorporation of GTP-bound tubulins from the solution (Vemu et al., 2018). The repaired site containing GTP-tubulins, in contrast to surrounding GDP-tubulin regions, now served as a 'rescue' spot where a shrinking plus end stops and regrows. Whether katanin in vivo nibbles particular microtubule locations, such as crossovers, and promotes amplification of microtubule mass without severing remains to be studied in plant cells.

**Conflicts of interest.** The authors declare no conflict of interest.

**Financial support.** The Japan Society for the Promotion of Science (JSPS) grants 20K21424 to M.N., 18KK0195 to M.N. and T.H., and 20H03276 to T.H., and the Human Frontier Science Program to M.N. ITbM is supported by the World Premier International Research Center Initiative (WPI), Japan.

**Authorship contributions.** All authors participated in writing the text and preparing the figures.

**Data availability statement.** No new data or code is presented in this paper.

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
