## [Reviewer Report]

December 20, 2021

Dear Editor of Quantitative Plant Biology;

We are herewith submitting a review article, entitles “Finding a right place to cut: how katanin is targeted to cellular severing sites”. Katanin severs plant microtubules at several subcellular locations, and molecular mechanisms for the selective targeting are recently uncovered. We believe that it is the right time to review this topic, and to appeal to the plant cell biologists.

Sincerely,

Takashi Hashimoto and Masayoshi Nakamura

Corresponding authors

---

## [Reviewer Report]

*Comments to Author*: This is a timely review for the recent progress on the regulatory mechanisms of plant katanin by the authors who are the specialists of the research field. Authors summarize the regulatory mechanisms into three events: crossover, nucleation, and cell division. The review is overall insightful enough and well written in a concise manner. I have only a few comments. 

1. The mechanism by which katanin is recruited to the microtubule crossover sites is an important topic to understand the role of katanin in plants. One of the problems of this topic may be that the precise geometry of microtubule crossover sites has not been clarified at nano scale level. Authors depicted this in Figure2 as a highly bended microtubule over the preexisting microtubules. I think this is too exaggerated and far from realistic view of microtubules in vivo as shown in a huge number of TEM images since 1960’. I am afraid that this picture may mislead students to an incorrect view.

2. Regulation by ROP-RIC pathway (Lin et al. [30]) is mentioned at Outlook, but I think it would be better to discuss this in main text. Also, it might be interesting if authors discuss how katanin is activated in interphase by ROPs and mechano-signaling by referring recent papers, Eng et al. 2021 and Tang et al. 2021.

3. It might be better to touch the suppression of MT severing by MAP65 (Burkart and Dixit [7]) and the possible interaction with ABS6 (Li et al. 2020) somewhere in the manuscript.

4. “ATPase Associated with diverse cellular Activities (AAA ATPase)” is “ATPase Associated with diverse cellular Activities (AAA) ATPase”?

5. Line 177: phoragmoplast is phragmoplast?

---

## [Reviewer Report]

*Comments to Author*: Microtubule severing by katanin is an important mechanism for constructing and remodeling the microtubule cytoskeleton in eukaryotes. This review focuses on how katanin is targeted to specific sites in plant cells. This is an important topic because correct localization and timing of microtubule severing is vital to cell growth and division.

I enjoyed reading this review. It is well written and authoritatively covers the literature in a balanced manner. There are a few minor points that should be addressed to improve this article:

1) In Figure 1, I recommend changing the icon for microtubules from crossed microtubules to a single microtubule.

2) In discussing why katanin seldom severs microtubules along their entire length in cells (lines 86-88), the authors focus on targeting of katanin to specific sites of action by the p80 subunit. A complementary mechanism is the protection of the microtubule lattice against severing by MAPs. Examples of MAPs that serve this function include tau in mammals (Qiang et al 2006 J of Neuroscience, and Siahaan et al 2019 Nature Cell Bio) and MAP65 in plants (Burkart et al [7] Mol Biology of the Cell). The authors should include this point in this section of the review.

3) On line 123, change severer to “more sever”

4) The authors should include the Leong et al [29] Cell Structure and Function paper on line 128-129. Similarly, they should include the Fan et al [15] Current Biology paper on lines 131 and 133.